# Equivariance with Learned Canonicalization Functions

**Sékou-Oumar Kaba**[†]                       KABASEKO@MILA.QUEBEC
*School of Computer Science, McGill University*
*Mila - Quebec Artficial Intelligence Institute*

**Arnab Kumar Mondal**[†]                ARNAB.MONDAL@MILA.QUEBEC
*School of Computer Science, McGill University*
*Mila - Quebec Artficial Intelligence Institute*

**Yan Zhang**                               YAN.ZHANG@MILA.QUEBEC
*Samsung - SAIT AI Lab, Montréal*

**Yoshua Bengio**                     YOSHUA.BENGIO@MILA.QUEBEC
*DIRO, Université de Montréal*
*Mila - Quebec Artficial Intelligence Institute*

**Siamak Ravanbakhsh**              SIAMAK@CS.MCGILL.CA
*School of Computer Science, McGill University*
*Mila - Quebec Artficial Intelligence Institute*

[†]denotes equal contribution

**Editors:** Sophia Sanborn, Christian Shewmake, Simone Azeglio, Arianna Di Bernardo, Nina Miolane

## Abstract

Symmetry-based neural networks often constrain the architecture in order to achieve invariance or equivariance to a group of transformations. In this paper, we propose an alternative that avoids this architectural constraint by learning to produce a canonical representation of the data. These canonicalization functions can readily be plugged into non-equivariant backbone architectures. We offer explicit ways to implement them for many groups of interest. We show that this approach enjoys universality while providing interpretable insights. Our main hypothesis is that learning a neural network to perform canonicalization is better than doing it using predefined heuristics. Our results show that learning the canonicalization function indeed leads to better results and that the approach achieves great performance in practice.

**Keywords:** deep learning, symmetry, equivariance, shape recognition, group theory, vision, cognition

## 1. Introduction

The problem of designing machine learning models that properly exploit the structure and symmetry of the data is becoming more important as the field is broadening its scope to more complex problems. In multiple applications, the transformations with respect to which we require a model to be *invariant* or *equivariant* are known and provide a strong inductive bias (Bronstein et al., 2021; Bogatskiy et al., 2022; van der Pol et al., 2020; Mondal et al., 2020; Celledoni et al., 2021).

As is often the case, taking a step back and drawing analogies with human cognition is fruitful here. Human pattern recognition handles some symmetries with relative ease. When data is transformed in a way that preserves its essential characteristics, we precisely know

if and how we should adapt our response. One context in which this has been particularly well-studied in cognitive science is visual shape recognition. Experiments have shown that subjects can accurately make the difference between different orientations of an object and actual modifications to the structure of an object (Shepard and Metzler, 1971; Carpenter and Eisenberg, 1978).

There are multiple ways in which this could be achieved. According to Tarr and Pinker (1989), theories of invariant shape recognition broadly fall into three categories: *viewpoint-independent* models, in which object representations depend only on invariants features, *multiple-view* models in which an object is represented as a set of representations corresponding to different orientations, and *single-view-plus-transformation* models in which an object is converted to a canonical orientation by a transformation process.

Correspondingly, similar ideas have been explored in deep learning to achieve equivariance. Models that impose equivariance through constraints in the architecture (Shawe-Taylor, 1989; Cohen and Welling, 2016; Ravanbakhsh et al., 2017) or that only use invariants as inputs (Villar et al., 2021) can be seen as belonging to the viewpoint-independent type. The multiple-view approach includes models that symmetrize the input by averaging over all the transformations or a subset of them (Manay et al., 2006; Benton et al., 2020; Yarotsky, 2022; Puny et al., 2022). By contrast, the transformation approach has seen less interest, and to our knowledge has not yet been used to achieve exact equivariance. This is all the more surprising considering that evidence from cognitive science suggests that this approach is used in human visual cognition (Shepard and Metzler, 1971; Carpenter and Eisenberg, 1978; Hinton and Parsons, 1981). When presented with a rotated version of an original pattern, the time taken by humans to do the association is proportional to the angle of rotation, which is more consistent with the hypothesis that we perform a *mental rotation*.

Some works (Jaderberg et al., 2015; Qi et al., 2017) have proposed to learn transformations of input to facilitate processing in a downstream task, but these approaches are closer to regularizers and provide no guarantees. Another method is to use heuristics to standardize inputs (Yüceer and Oflazer, 1993; Lowe, 2004; Aslan et al., 2022), but this approach requires significant hand-engineering and is hardly generalizable.

**Present work** We introduce a systematic and general method for equivariant machine learning based on learning mappings to canonical samples. We hypothesize that among all valid canonicalization functions, some will lead to better downstream performance than others. Rather than trying to hand-engineer these functions, we may as well let them be learned in an end-to-end fashion with a prediction neural network. Our method can readily be used as an independent module that can be plugged into existing architectures to make them equivariant to arbitrary transformation groups, discrete or continuous. Our approach enjoys similar expressivity advantages to methods like *frame averaging* by (Puny et al., 2022), but has several added benefits. It is simpler, more efficient, and replaces hand-engineered frames for each group by a systematic end-to-end learning approach.

Our contributions are as follow:

- **Novel Framework:** We introduce a general framework for equivariance to arbitrary groups based on mappings to canonical samples. This framework can be plugged into any existing non-equivariant architecture.

- **Theoretical Guarantees:** We prove that in some settings, such models are universal approximators of equivariant functions.

- **Efficient Implementations:** We provide multiple variants of efficient implementations of this framework to specific domains.

- **Practical Performance:** We perform experiments that show that the proposed method achieves excellent results in the image domain. We also support our hypothesis that learning the canonicalization function is a better strategy than fixing it.

## 2. Canonicalization Functions

### 2.1. Problem Setting

We are interested in learning functions $\phi : \mathcal{X} \to \mathcal{Y}$ with inputs $\mathbf{x} \in \mathcal{X}$ and outputs $\mathbf{y} \in \mathcal{Y}$ belonging to a arbitrary finite-dimensional vector spaces. We will consider a set of linear symmetry transformations $T \subset \mathrm{GL}\left(\mathcal{X}\right)$, where $\mathrm{GL}\left(\mathcal{X}\right)$ is the set of invertible matrices over the vector space $\mathcal{X}$. This is described by a group representation $\rho : G \to T$, where $G$ is an abstract group. Without loss of generality, we can assume that $\rho$ is a group isomorphism. Therefore, the inverse $\rho^{-1} : T \to G$ is defined. A function $\phi$ is $G$-equivariant if

$$\phi\left(\rho\left(g\right)\mathbf{x}\right) = \rho'\left(g\right)\phi\left(\mathbf{x}\right), \ \forall \ g, \mathbf{x} \in G \times \mathcal{X}, \tag{1}$$

where the group action $\rho$ on the input and the group action $\rho'$ on the output will be clear from the context. In particular, when $\rho'\left(g\right) = I$, we say that $\phi$ is invariant. We use $\rho\left(G\right)$ to denote the image of $\rho$.

We call $\rho\left(G\right)\mathbf{x} = \{\rho\left(g\right)\mathbf{x} \mid \forall \ g \in G\}$ the orbit of the element $\mathbf{x}$. It is the set of elements to which $\mathbf{x}$ can be mapped to by the group action. The set of orbits, denoted $\mathcal{X}/G$ forms a partition of the set $\mathcal{X}$.

### 2.2. General Formulation

The invariance requirement on a function $\phi$ amounts to having all the members of a group orbit mapped to the same image by $\phi$. It is thus possible to achieve invariance by appropriately mapping all elements to a canonical sample from their orbit before applying any function. For equivariance, elements can be mapped to a canonical sample and, after a function is applied, transformed back according to their original position in the orbit. This can be formalized by writing the equivariant function $\phi$ in *canonicalized form* as

$$\phi\left(\mathbf{x}\right) = h'\left(\mathbf{x}\right)\mathbf{f}\left(h\left(\mathbf{x}\right)^{-1}\mathbf{x}\right) \tag{2}$$

where the function $\mathbf{f} : \mathcal{X} \to \mathcal{Y}$ is called the prediction function and the function $h : \mathcal{X} \to \rho\left(G\right)$ is called the canonicalization function. Here $h\left(\mathbf{x}\right)^{-1}$ is the inverse of the representation matrix and $h'\left(\mathbf{x}\right) = \rho'\left(\rho^{-1}\left(h\left(\mathbf{x}\right)\right)\right)$ is the counterpart of $h\left(\mathbf{x}\right)$ on the output.

Equivariance in equation 2 is obtained for any prediction function if the canonicalization function is itself $G$-equivariant[1], $h\left(\rho\left(g\right)\mathbf{x}\right) = \rho\left(g\right)h\left(\mathbf{x}\right) \ \forall \ g, x \in G \times \mathcal{X}$.

---

1. Symmetric inputs in $\mathcal{X}$ pose a problem if we use the standard definition of equivariance for the canonicalization function. We explain this in Appendix A and we introduce a relaxed version of equivariance that

It may seem like the problem of obtaining an equivariant function has merely been transferred in this formulation. This is however not the case: in equation 2, the equivariance and prediction components are effectively decoupled. The canonicalization function $h$ can therefore be chosen as a simple and inexpressive equivariant function, while the heavy-lifting is done by the prediction function $\mathbf{f}$.

A more general condition can be formulated, such that the decoupling is partial.

**Theorem 1** *For some subgroup $K \leq G$, if $\forall\ g, \mathbf{x} \in G \times \mathcal{X}$ there exists a $k \in K$ such that*

$$h\left(\rho\left(g\right)\mathbf{x}\right) = \rho\left(g\right) h\left(\mathbf{x}\right) \rho\left(k\right) \tag{3}$$

*and the prediction function $\mathbf{f}$ is $K$-equivariant, then $\phi$ defined in equation 2 is $G$-equivariant. If $K$ is a normal subgroup such that $G \simeq J \ltimes K$, this can be realized with a canonicalization function that has image $\rho\left(J\right)$, and that is $J$-equivariant and $K$-invariant.*

The proof follows in Appendix B. This is equivalent to saying that the canonicalization function should output a representation of a coset in $G/K$ in an equivariant way, the applied transformation being chosen arbitrarily within the coset.

When $K = \{e\}$, only the canonicalization function is constrained, which is the case described above. The other extreme, given by $K = G$, corresponds to constraining the prediction function as is usually done in equivariant architectures. These are respectively the *single-view-plus-transformation* and the *viewpoint-independent* implementations described in the introduction. Subgroups $\{e\} < K < G$ offer intermediary options; the lattice of subgroups of $G$ therefore defines a family of models. Since equivariance to a smaller group is less constraining for the prediction function, set inclusion in the subgroup lattice corresponds to increased expressivity.

### 2.3. Universality Result

We can now introduce a more formal result on the universality of equivariant functions obtained with canonicalization functions. A parameterized function $\phi$ is a universal approximator of $G$-equivariant functions if for any $G$-equivariant continuous function $\psi$, any compact set $\mathcal{K} \subseteq \mathcal{X}$ and any $\epsilon > 0$, there exists a choice of parameters such that $\|\psi\left(\mathbf{x}\right) - \phi\left(\mathbf{x}\right)\| < \epsilon \ \forall\ \mathbf{x} \in \mathcal{K}$. We make the additional assumption that the set $\mathcal{K}$ is closed under the group action.

**Theorem 2** *A $G$-equivariant parameterized function $\phi$ written in canonicalized form (equation 2) and satisfying equation 3 with $K \leq G$, is a universal approximator of $G$-equivariant functions if the prediction function is a universal approximator of $K$-equivariant functions.*

The proof follows in Appendix C. The following corollary is especially relevant.

**Corollary 3** *A $G$-equivariant parameterized function $\phi$ written in canonicalized form with a $G$-equivariant canonicalization function and a multilayer perceptron (MLP) as a prediction function is a universal approximator of $G$-equivariant functions.*

This result can significantly simplify the design of universal approximators of equivariant functions since a non-universal equivariant architecture for the canonicalization function can be combined with an MLP.

---

solves this issue. However, because the subset of symmetric inputs is of measure zero, using standard equivariance is not expected to be a problem in practice.

## 3. Model Design for Canonicalization Functions

In the next section, we elaborate on how suitable canonicalization functions can be obtained in different settings.

### 3.1. Euclidean Group

The Euclidean group $E(d)$ is used to capture symmetry to rotations, translations, and reflections. Domains in which this type of symmetry is especially relevant include point cloud modeling, image processing and applications in physics. Below we give the design principles to get an equivariant model for image and point cloud inputs.

**Image Input.** In order to build networks equivariant to $E(2)$, we need a canonicalization function that outputs an element of $E(2)$ given an image. This can be achieved by using a G-CNN (Cohen and Welling, 2016) and taking an argmax of the output feature map. While G-CNN only considers four rotations this can be easily extended to any arbitrary rotations. This approach can be further simplified if we are using a translation equivariant prediction network which is generally the case with most CNN-based architectures. As the translation group $T(2)$ is a normal subgroup of the Euclidean group $E(2)$, Theorem 1 indicates that we only require the canonicalization function to be equivariant to $O(2)$. This means we can eliminate the spatial dimension in the output feature map of the canonicalization function and only need to take an argmax along the point group dimension to identify the correct orientation of the image. However, there are two problems with this approach. First, extending G-CNNs to higher-order rotations will lead to more computation and artifacts due to the finer rotation of filters. Second, we can not backpropagate through the canonicalization function as the argmax operation is not differentiable.

We can avoid the first problem by using a shallower network with a larger filter size. We empirically show why this is a sound choice for canonicalization function in Section 4. Furthermore to solve the second problem we can use the straight-through estimator trick (Bengio et al., 2013). Appendix E contains a PyTorch code snippet to perform the canonicalization function of images in a differentiable way using a G-CNN. Similarly, the output of the canonicalization function can be used to invert the feature maps from the prediction network back to their original orientation in a differentiable way.

Since CNNs are universal approximators of $T(2)$-equivariant functions (Yarotsky, 2022), it follows from Theorem 2 that a CNN augmented with an $O(2)$ equivariant canonicalization function is a universal approximator of $E(2)$-equivariant functions.

**Point Cloud Input.** Elements of the Euclidean group can be written as $(\mathbf{O}, \mathbf{t})$, where $\mathbf{O} \in \mathbb{R}^{n \times n}$ is an orthogonal matrix and $\mathbf{t} \in \mathbb{R}^n$ is an arbitrary translation vector. The vector representation on $n+1$ dimensional vectors (defined by concatenating a constant 1 to the original vectors) is then defined in the following way

$$\rho(\mathbf{O}, \mathbf{t}) = \begin{pmatrix} \mathbf{O} & \mathbf{t} \\ \mathbf{t}^T & 1 \end{pmatrix} \tag{4}$$

This representation is bounded We seek to define an $E(d)$-equivariant canonicalization function. This can be done by defining it as $h(\mathbf{x}) = \rho\big(h^O(\mathbf{x}), h^t(\mathbf{x})\big)$, where the functions

$h^O : \mathcal{X} \rightarrow \mathbb{R}^{n \times n}$ and $h^t : \mathcal{X} \rightarrow \mathbb{R}^n$ output the point transformation and the translation respectively. By the equivariance condition,

$$\left(h^O\left(\rho(\mathbf{O}, \mathbf{t})\mathbf{x}\right), h^t\left(\rho(\mathbf{O}, \mathbf{t})\mathbf{x}\right)\right) = \left(\mathbf{O}h^O\left(\mathbf{x}\right), \mathbf{O}h^t\left(\mathbf{x}\right) + \mathbf{t}\right) \tag{5}$$

which means that $h^O$ must be $O(d)$-equivariant and translation invariant, and that $h^t$ must be $E(d)$-equivariant. These constraints can be satisfied by using already existing equivariant architectures. Since most of the work will be done by a prediction function that can be very expressive, like Pointnet (Qi et al., 2017), a simple and efficient architecture can be used for the canonicalization function, for example, Vector Neurons (Deng et al., 2021) or $E(d)$-Equivariant Graph Neural Networks (Satorras et al., 2021). The output of $h^O$ can be made an orthogonal matrix by having it output $n$ vectors and orthonormalizing them with the Gram-Schmidt procedure, which is itself equivariant.

Using Deep Sets (Zaheer et al., 2017) as a backbone architecture would result in a universal approximator of $E(d)$ and permutation equivariant functions, following Theorem 2 and Theorem 1 of (Segol and Lipman, 2020).

### 3.2. Symmetric Group

The symmetric group $S_n$ over a finite set of $n$ elements contains all the permutations of that set. This group captures the inductive bias that input order should not matter. Domains for which $S_n$-equivariance is desirable include object modelling and detection, graph representation learning, and applications in language modeling.

$S_n$-equivariant canonicalization functions can be obtained with a direct approach. In particular, the methods introduced by Mena et al. (2018); Cuturi et al. (2019); Zhang et al. (2019) can be used to obtain the permutation of the input. However, these approaches only learn relaxations of permutations and not actual elements of the permutation group. In addition, they offer no clear way to handle sets of different sizes.

We propose an alternative of $S_n$-equivariance for the canonicalization function that circumvents these issues. Instead of producing a permutation of the input, the canonicalization function can instead be designed to obtain a clustering of the input, with a function $c : \mathcal{X} \rightarrow \{1, \ldots, k\}$, where $k$ is the number of possible clusters. This can be made differentiable using the straight-through estimator described for the images above. Then, a corresponding permutation is obtained by sorting elements with respect to their cluster index, with elements belonging to the same cluster being sorted arbitrarily. The canonicalization function would therefore respect equation 3 with respect to the subgroup $K = S_{l_1} \times \cdots \times S_{l_k}$, where $l_i$ is the number of elements assigned to cluster $i$, the product of symmetric groups within clusters. Heuristically, this enforces the inductive bias that set elements should be mapped to high-level classes, across which interactions will be evaluated in an expressive way. This is close in spirit to what is done in the ClusterFormer (Wang et al., 2022) and Reformer (Kitaev et al., 2020) transformer architectures.

The prediction function $\mathbf{f}$ would then have to be $S_{l_1} \times \cdots \times S_{l_k}$ equivariant to obtain permutation equivariance. Such a function can readily be obtained with an equivariant multilayer perceptron, as derived by Wang et al. (2020).

## 4. Experiments

We perform a preliminary empirical analysis of the proposed framework in the image domain. Experiments on other domains will be the subject of more extensive work. We selected the Rotated MNIST dataset (Larochelle et al., 2007) which is used as a benchmark dataset that uses a classification task to test equivariant architectures in prior work (Cohen and Welling, 2016). In all our experiments we only change the network architecture. We train the models by minimizing the cross entropy loss for 100 epochs using Adam (Kingma and Ba, 2014) with a learning rate of 0.001. We perform early stopping based on the classification performance of the validation dataset with patience of 20 epochs.

| Method | Error % ↓ |
|---|---|
| CNN (base) | 4.90 ±0.20 |
| G-CNN (p4) | 2.28 ±0.00 |
| G-CNN (p4 & = params) | 2.36 ±0.15 |
| CNN (= params) | 4.80 ±0.37 |
| CNN (w PCA) | 3.35 ±0.21 |
| Ours (p4 & frozen) | 3.91 ±0.12 |
| Ours (p4) | 2.41 ±0.10 |

Table 1: Comparison with the existing work for Rotated-MNIST.

We first compare our method with different CNN and G-CNN baselines. For CNN (base), we choose an architecture with 7 layers where layer 1 to 3 has 32, 4 to 6 has 64 and layer 7 has 128 channels respectively. Instead of pooling, we use convolution filters of size $5 \times 5$ with a stride 2 at layers 4 and 7. The remaining convolutions have filters of size $3 \times 3$ and stride 1. All the layers are followed by batch-norm and ReLU activation with dropout(p=0.4) only at layers 4 and 7. For the canonicalization function, we choose a shallow G-CNN with three layers. We start with a lifting layer with a filter that is the same size as the input image. This is followed by ReLU and group equivariant layers with $1 \times 1$ filters.

We consider two variants: an untrained canonicalization function with frozen weights and a canonicalization function learned end-to-end with the prediction function. For a pure G-CNN based baseline, we provide the value reported by Cohen and Welling (2016) and design a variant which has similar architecture to CNN (base) while matching the number of parameters of our model with canonicalization function. We call this G-CNN (p4 & =params). We consider two more variants of the CNN: one with the same number of parameters as our model and the other where the canonicalization function is done by finding the orientation of the digits using Principal Component Analysis (PCA).

We see that using a fixed canonicalization function technique like PCA or canonicalization function with frozen parameters leads to an improvement in performance over the CNN baseline. However, learning canonicalization function provides a significant performance improvement. Our approach outperforms all the CNN-based baselines and is comparable to G-CNNs. We further notice that adding canonicalization function for reflections hurts performance. This is in fact not surprising as reflection augmentations were not used to build the rotated-MNIST dataset. Adding reflection invariance while the data does not possess it may introduce ambiguity between certain digits like 2 and 5. Next, we perform experiments to understand the role of different components in our model using the group of $n$ discrete rotations $(pn)$.

### 4.1. Ablation Study

First, we vary both the number of layers of the canonicalization network and the number of rotations it is equivariant to. For this, we extend the layers of GCNN to any arbitrary rotations. As we noticed that using a larger filter leads to better performance for higher order rotations we stick to architecture with a lifting layer that has image-sized filters followed by $1 \times 1$ filters. From Table 4.1, we notice that adding equivariance to higher order rotation in the canonicalization function leads to significant performance improvement in comparison to adding more layers. Figure 1 shows the canonical orientation resulting from the learnt canonicalization function with a single lifting layer on 110 randomly sampled images of class 7 from the test dataset. This suggests that having a shallow network is enough to learn the correct canonicalization function with a sufficiently high order of discrete rotations. For p64, we see that all the similar-looking samples are aligned in one particular orientation. In contrast to this, although techniques like PCA or freezing parameters of the canonicalization function finds the correct canonicalization function for simple digits like 1 (see Appendix 3) they struggle to find stable mappings for more complicated digits like 7.

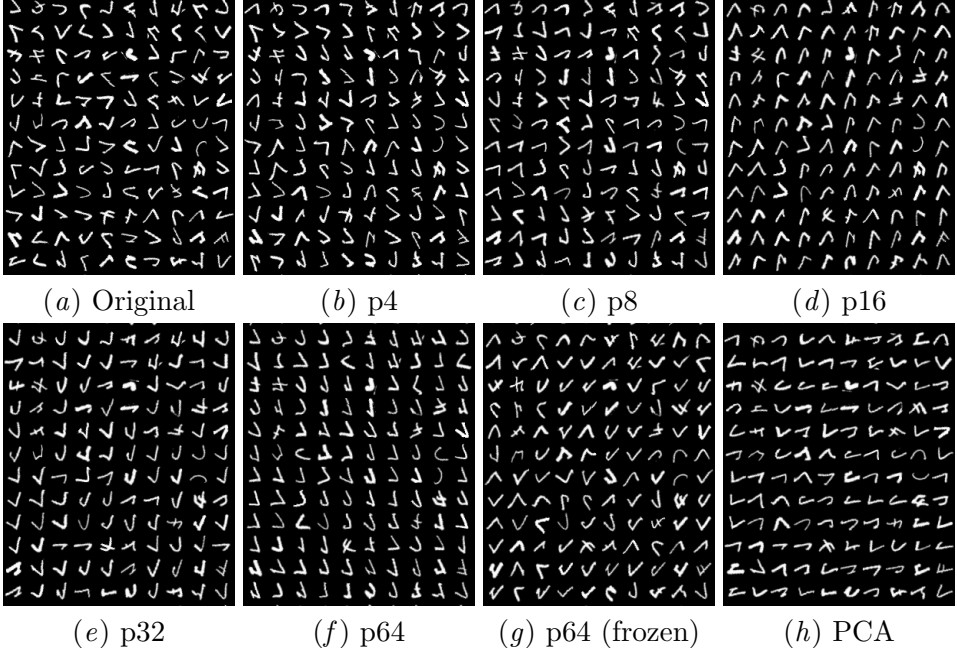

Figure 1: Canonicalized images from different canonicalization functions for digit 7.

Next, we compare the inference time of our model with pure G-CNN-based architectures. For this experiment, we take the CNN architecture of our predictor network and replace the convolutions with group convolutions. As increasing the order of rotation in G-CNN requires more copies of rotated filters in the lifting layer and more parameters in the subsequent group convolution layers, we decreased the number of channels to keep the number of parameters the same as our model. Figure 2 shows that although G-CNN is slightly better for the p4 group increasing the order of discrete rotations improves our model's performance

| Num layers | Order of the discrete rotation group | | | | |
|---|---|---|---|---|---|
| | p4 | p8 | p16 | p32 | p64 |
| 1 | 2.52 ±0.12 | 2.37 ±0.09 | 2.20 ±0.08 | 2.05 ±0.15 | 2.01 ±0.09 |
| 2 | 2.44 ±0.06 | 2.31 ±0.05 | 2.16 ±0.09 | 2.00 ±0.07 | 2.02 ±0.12 |
| 3 | 2.41 ±0.11 | 2.28 ±0.09 | 2.11 ±0.06 | 1.98 ±0.09 | 1.99 ±0.10 |

Table 2: Impact of the number of layers in canonicalization function network and order of the discrete rotations to which it is equivariant on the performance.

significantly compared to G-CNN. In addition to performance gain, our model's inference speed remains more or less constant while encoding invariance to higher-order rotations due to the shallow canonicalization network. This makes our approach more suitable for building equivariance to bigger groups.

## 5. Conclusion

In this work, we have proposed to use learned canonicalization function to obtain equivariant machine learning models. These canonicalization functions can conveniently be plugged into existing architectures, resulting in highly expressive models. We have described general approaches to obtain canonicalization functions and specific implementations for the Euclidean group (for images and point clouds) and for the symmetric group.

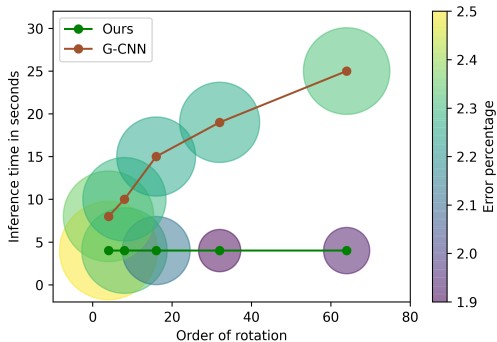

Figure 2: Inference time comparison of our method with G-CNN with increasing order of rotations.

We performed preliminary experimental studies in the image domain to test our initial hypotheses regarding this framework. First, we show that our approach achieves comparable or better performance than baselines on invariant tasks. Importantly, learning the canonical network is a better approach than using a fixed mapping, either a frozen neural network or a heuristic approach like PCA. Our results also show that the canonicalization function can be realized with a shallow equivariant network, without hindering performance. Finally, we show that this approach reduces inference time and is more suitable for bigger groups compared to G-CNNs.

Areas of future work include an experimental study of this framework on other domains, such as point clouds and graphs. In the image domain, we will also explore building canonicalization functions using steerable networks. The function would output an orientation fibre that transforms by the irreducible representation of the special orthogonal group. Understanding how design choices for canonicalization functions affect downstream performance would also be a potentially fruitful research direction.

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

## Appendix A. Symmetric inputs and relaxed equivariance

We say that an input $\mathbf{x} \in \mathcal{X}$ is symmetric if its stabilizer subgroup $G_{\mathbf{x}} = \{g \in G \mid \rho(g)\mathbf{x} = \mathbf{x}\}$ is non-trivial.

Given any $g_1, g_2 \in G$, a necessary and sufficient condition for

$$\rho(g_1)\mathbf{x} = \rho(g_2)\mathbf{x} \tag{6}$$

is that $g_1$ and $g_2$ are part of the same coset for the stabilizer, e.g. $g_1, g_2 \in gG_{\mathbf{x}}$. This is follows from the well-known relation between orbits and stabilizers. Therefore, symmetric inputs are always fixed by multiple group elements.

Symmetric inputs are problematic when using the standard definition of equivariance for the canonicalization function because for $g_1, g_2 \in gG_{\mathbf{x}}$, we have

$$h(\rho(g_1)\mathbf{x}) = h(\rho(g_2)\mathbf{x}) \tag{7}$$
$$\rho'(g_1)h(\mathbf{x}) = \rho'(g_2)h(\mathbf{x}) \tag{8}$$

For general group actions, there will not exist a $h(\mathbf{x}) \in \rho(G)$ such that the last equality is satisfied.

A relaxed version of equivariance can be defined to address this issue.

**Definition 4 (Relaxed equivariance)** *Given group representations $\rho : G \to \mathrm{GL}(\mathcal{X})$ and $\rho' : G \to \mathrm{GL}(\mathcal{Y})$, a function $h : \mathcal{X} \to \mathcal{Y}$ satisfies the relaxed equivariance condition if $\forall g_1, \mathbf{x} \in G \times X$ there exists a $g_2 \in g_1 G_{\mathbf{x}}$ such that*

$$h(\rho(g_1)\mathbf{x}) = \rho'(g_2)h(\mathbf{x}) \tag{9}$$

This is a generalization of the idea of multiset-equivariance introduced by [Zhang et al. (2022)](#) to arbitrary group representations. When $G_{\mathbf{x}} = \{e\}$, standard equivariance is recovered. Canonicalization functions satisfying this condition do not suffer from the aforementioned problem. In addition, this condition is sufficient to obtain relaxed equivariance for canonicalized functions (Eq. [2](#)).

This is because, for $g_2 \in g_1 G_{\mathbf{x}}$ :

$$\phi(\rho(g_1)\mathbf{x}) = h'(\rho(g_1)\mathbf{x})\mathbf{f}\left(h(\rho(g_1)\mathbf{x})^{-1}\rho(g_1)\mathbf{x}\right) \tag{10}$$

$$\phi(\rho(g_1)\mathbf{x}) = \rho'(g_2)h'(\mathbf{x})\mathbf{f}\left(h(\mathbf{x})^{-1}\rho(g_2)^{-1}\rho(g_1)\mathbf{x}\right) \tag{11}$$

$$\phi(\rho(g_1)\mathbf{x}) = \rho'(g_2)h'(\mathbf{x})\mathbf{f}\left(h(\mathbf{x})^{-1}\rho(g_2^{-1}g_1)\mathbf{x}\right) \tag{12}$$

Using the fact that $g_2^{-1}g_1 \in G_{\mathbf{x}}$,

$$\phi(\rho(g_1)\mathbf{x}) = \rho'(g_2)h'(\mathbf{x})\mathbf{f}\left(h(\mathbf{x})^{-1}\mathbf{x}\right) \tag{13}$$

$$\phi(\rho(g_1)\mathbf{x}) = \rho'(g_2)\phi(\mathbf{x}) \tag{14}$$

The fact that we obtain only relaxed equivariance for the canonicalized function is a feature rather than a bug. This captures the desideratum that a function should be able to output asymmetric outputs from symmetric inputs, which is not the case for standard equivariant functions.

## Appendix B. Proof of Theorem 1

We first show equivariance for a general subgroup $K$. We start with

$$\phi\left(\rho\left(g\right)\mathbf{x}\right) = h'\left(\rho\left(g\right)\mathbf{x}\right)\mathbf{f}\left(h\left(\rho\left(g\right)\mathbf{x}\right)^{-1}\rho\left(g\right)\mathbf{x}\right) \tag{15}$$

If equation 3 is satisfied, then $\forall\, g, \mathbf{x} \in G \times \mathcal{X}$ there is a $k \in K$ such that

$$\phi\left(\rho\left(g\right)\mathbf{x}\right) = \rho'\left(g\right)h'\left(\mathbf{x}\right)\rho'\left(k\right)\mathbf{f}\left(\left[\rho\left(g\right)h\left(\mathbf{x}\right)\rho\left(k\right)^{-1}\right]^{-1}\rho\left(g\right)\mathbf{x}\right) \tag{16}$$

$$\phi\left(\rho\left(g\right)\mathbf{x}\right) = \rho'\left(g\right)h'\left(\mathbf{x}\right)\rho'\left(k\right)\mathbf{f}\left(\rho\left(k\right)^{-1}h\left(\mathbf{x}\right)^{-1}\rho\left(g\right)^{-1}\rho\left(g\right)\mathbf{x}\right) \tag{17}$$

Using the $K$-equivariance of $\mathbf{f}$, we obtain

$$\phi\left(\rho\left(g\right)\mathbf{x}\right) = \rho'\left(g\right)h'\left(\mathbf{x}\right)\rho'\left(k\right)\rho'\left(k\right)^{-1}\mathbf{f}\left(h\left(\mathbf{x}\right)^{-1}\mathbf{x}\right) \tag{18}$$

$$\phi\left(\rho\left(g\right)\mathbf{x}\right) = \rho'\left(g\right)h'\left(\mathbf{x}\right)\mathbf{f}\left(h\left(\mathbf{x}\right)^{-1}\mathbf{x}\right) \tag{19}$$

Now, consider the special case where $K$ is a normal subgroup of $G$ such that the group can be taken to be isomorphic to a semidirect product $G \simeq K \rtimes J$. Then, group elements can be written as $g = (k, j)$, where $k \in K$ and $j \in J$. The product is defined as $g_1 g_2 = (k_1, j_1)(k_2, j_2) = (k_1 \varphi[j_1](k_2), j_1 j_2)$, where $\varphi : J \to \mathrm{Aut}(K)$ is a group homomorphism. Setting $k_2 = e$ and $j_1 = e$, we get any group element as $(k_1, e)(e, j_2) = (k_1, j_2)$.

If the canonicalization function is $J$-equivariant and $K$-invariant, we have

$$h\left(\rho\left(k, j\right)\mathbf{x}\right) = h\left(\rho\left(k, e\right)\rho\left(e, j\right)\mathbf{x}\right) \tag{20}$$

$$h\left(\rho\left(k, j\right)\mathbf{x}\right) = \rho\left(e, j\right)h\left(\mathbf{x}\right) \tag{21}$$

We then show that there is a $k' \in K$ such that equation 3 is satisfied. Multiplying by $\rho(e) = \rho(k, e)\rho(e, j)h(\mathbf{x})h(\mathbf{x})^{-1}\rho(e, j)^{-1}\rho(k, e)^{-1}$ on the left, we have

$$\rho\left(e, j\right)h\left(\mathbf{x}\right) = \rho\left(k, e\right)\rho\left(e, j\right)h\left(\mathbf{x}\right)h\left(\mathbf{x}\right)^{-1}\rho\left(e, j\right)^{-1}\rho\left(k, e\right)^{-1}\rho\left(e, j\right)h\left(\mathbf{x}\right) \tag{22}$$

Using the fact that conjugation of an element of $K$ by an element of $G$ preserves $K$ membership, we define $\rho(k', e) = h(\mathbf{x})^{-1}\rho(e, j)^{-1}\rho(k, e)^{-1}\rho(e, j)h(\mathbf{x})$

$$\rho\left(e, j\right)h\left(\mathbf{x}\right) = \rho\left(k, e\right)\rho\left(e, j\right)h\left(\mathbf{x}\right)\rho\left(k', e\right) \tag{23}$$

which shows that equation 3 is satisfied.

Finally, we show that in this case, the image of $h$ can be chosen to be $\rho(J)$. We first remark that in each orbit $\mathcal{X}/G$ of the group action, the canonical sample $\hat{\mathbf{x}}$ can be obtained from any orbit member $\mathbf{x}$, as $\hat{\mathbf{x}} = h(\mathbf{x})^{-1}\mathbf{x}$. For the canonical sample, we must have a $k \in K$ such that

$$h\left(h\left(\mathbf{x}\right)^{-1}\mathbf{x}\right) = h\left(\mathbf{x}\right)^{-1}h\left(\mathbf{x}\right)\rho\left(k, e\right) \tag{24}$$

If we impose $k = e$ to satisfy this condition, we have $h(\hat{\mathbf{x}}) = \rho(e, e)$.

Since any orbit member can conversely be written as $\mathbf{x} = \rho\left(k, j\right)\hat{\mathbf{x}}$ for some $k \in K$ and $j \in J$, if the canonicalization function is $J$-equivariant and $K$-invariant, we have

$$h\left(\mathbf{x}\right) = h\left(\rho\left(k, j\right)\hat{\mathbf{x}}\right) \tag{25}$$

$$h\left(\mathbf{x}\right) = \rho\left(e, j\right)h\left(\hat{\mathbf{x}}\right) \tag{26}$$

$$h\left(\mathbf{x}\right) = \rho\left(e, j\right) \tag{27}$$

which completes the proof.

## Appendix C. Proof of Theorem 2

**Proof** The proof is inspired by the symmetrization approach of Yarotsky (2022).

Let $\boldsymbol{\psi}$ be an arbitrary $G$-equivariant function, and $\boldsymbol{\phi}$ be defined by equation 2. By the equivariance of $\boldsymbol{\psi}$, we have

$$\left\|\boldsymbol{\psi}\left(\mathbf{x}\right) - \boldsymbol{\phi}\left(\mathbf{x}\right)\right\| = \left\|h'\left(\mathbf{x}\right)\boldsymbol{\psi}\left(h\left(\mathbf{x}\right)^{-1}\mathbf{x}\right) - h'\left(\mathbf{x}\right)\mathbf{f}\left(h\left(\mathbf{x}\right)^{-1}\mathbf{x}\right)\right\| \tag{28}$$

Since $\mathcal{Y}$ is finite-dimensional, we know that linear operators in $\mathrm{GL}\left(\mathcal{Y}\right)$ are bounded. This means that for every representation matrix there exists a positive number $c$ that bounds the induced operator norm, e.g. $\forall g \in G,\ \exists c > 0,\ \left\|\rho'\left(g\right)\right\| \leq c$.

We therefore obtain

$$\left\|\boldsymbol{\psi}\left(\mathbf{x}\right) - \boldsymbol{\phi}\left(\mathbf{x}\right)\right\| \leq \left\|h\left(\mathbf{x}\right)\right\|\left\|\boldsymbol{\psi}\left(h\left(\mathbf{x}\right)^{-1}\mathbf{x}\right) - \mathbf{f}\left(h\left(\mathbf{x}\right)^{-1}\mathbf{x}\right)\right\| \tag{29}$$

$$\left\|\boldsymbol{\psi}\left(\mathbf{x}\right) - \boldsymbol{\phi}\left(\mathbf{x}\right)\right\| \leq c\left\|\boldsymbol{\psi}\left(h\left(\mathbf{x}\right)^{-1}\mathbf{x}\right) - \mathbf{f}\left(h\left(\mathbf{x}\right)^{-1}\mathbf{x}\right)\right\| \tag{30}$$

where $c > 0$.

If $\mathbf{f}$ is a universal approximator of $K$-equivarant functions, then it is also a universal approximator of $G$-equivariant functions. We therefore have

$$\left\|\boldsymbol{\psi}\left(\mathbf{x}\right) - \mathbf{f}\left(\mathbf{x}\right)\right\| \leq \epsilon,\ \forall\,\mathbf{x} \in \mathcal{K} \tag{31}$$

Using the fact that $\mathcal{K}$ is closed under the group action, we obtain the desired result

$$\left\|\boldsymbol{\psi}\left(\mathbf{x}\right) - \boldsymbol{\phi}\left(\mathbf{x}\right)\right\| \leq \epsilon,\ \forall\,\mathbf{x} \in \mathcal{K} \tag{32}$$

■

**Appendix D. Additional results**

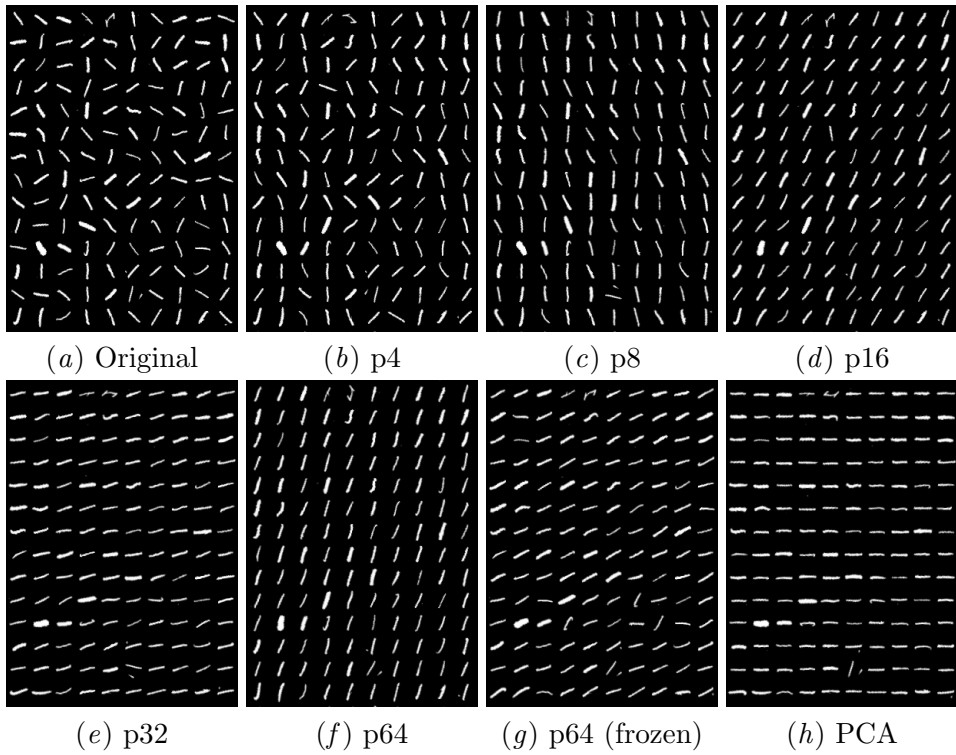

$(a)$ Original  $(b)$ p4  $(c)$ p8  $(d)$ p16

$(e)$ p32  $(f)$ p64  $(g)$ p64 (frozen)  $(h)$ PCA

Figure 3: Canonicalized images obtained from different canonicalization functions for digit 1.

## Appendix E.  Algorithm for Image Inputs

---

**Algorithm 1** Differentiable Canonicalization for Image Inputs

---

```python
import torch.nn.functional as F
import kornia as K

def get_canonicalized_images(images, fibre_features, use_reflection=True):
    """
    images: Tensor with shape (batch_size, in_channels, height, width)
    fibres_features: Tensor with shape: (batch_size, num_group_elements)
    use_reflection: Boolean
    :return: (batch_size, in_channels, height, width)
    """
    num_group_elements = fibre_features.shape[-1]
    num_rotations = num_group_elements // 2 if use_reflection else num_group_elements

    fibre_features_one_hot = F.one_hot(
        torch.argmax(fibre_features, dim=-1),
        num_group_elements
    ).float()

    fibre_features_soft = F.softmax(fibre_features, dim=-1)
    ref_angles = torch.linspace(0., 360., num_rotations+1)[:num_rotations]

    if use_reflection:
        ref_angles = torch.cat([ref_angles, ref_angles], dim=0)

    angles = torch.sum((
        fibre_features_one_hot + fibre_features_soft - fibre_features_soft.detach()
        ) * ref_angles, dim=-1)

    if use_reflection:
        reflect_one_hot = torch.cat(
            [torch.zeros(num_rotations), torch.ones(num_rotations)]
            , dim=0)
        reflect_indicator = torch.sum((
            fibre_features_one_hot + fibre_features_soft - fibre_features_soft.detach()
            ) * reflect_one_hot, dim=-1)

        images_reflected = K.geometry.hflip(images)
        reflect_indicator = reflect_indicator[:,None,None,None]
        images = (1 - reflect_indicator) * x + reflect_indicator * images_reflected

    return K.geometry.rotate(images, -angles)

# Use a shallow G-CNN as a canonicalization_network
feature_map = canonicalization_network(images)
# feature_map shape: (batch_size, num_channels, num_group_elements, height, width)

fibre_features = feature_map.mean(dim=(1, 3, 4))
# fibre_features shape: (batch_size, num_group_elements)

canonicalized_images = get_canonicalized_images(images, fibre_featuresr)
```

---

