# OpenReview forum: "Equivariance With Learned Canonicalization Functions"
_NeurIPS.cc/2022/Workshop/NeurReps — NeurReps 2022 Oral_

### Official Review · Reviewer_sU99 · 2022-10-07
**Simple elegant method for equivariance by picking a canonical frame. Concerns about symmetric inputs**

**Confidence:** 4
**Soundness:** 2
**Presentation:** 4
**Contribution:** 3
**Overall Rating:** 6

**Summary:**

The authors propose to build equivariant neural networks by having an equivariant mapping from the input to the group. With the inferred group element, the inputs are mapped to a canonical frame and a non-equivariant network can be used. Finally, the output is rotated back to the input frame, so that the resulting network is equivariant.

The authors also describe a variant in which the group is a semi-direct product, and the canonization is only wrt the point-group, and the applied network is only equivariant wrt the normal subgroup.

To canonize, the authors consider a direct method that uses an equivariant network, or alternatively a method in which the group element is found via optimizing a score function over all poses.

The method is evaluated on rotated MNIST.

**Questions:**

- It would be great if the authors could relate their method better to [1].
- The authors say that Gram-Schmidt equivariant, but I think it's not. I believe an equivariant orthogonal projection can be made by SVD via $M=USV^T \mapsto U V^T$

[1] Puny, Omri, Matan Atzmon, Heli Ben-Hamu, Ishan Misra, Aditya Grover, Edward J. Smith, and Yaron Lipman. 2021. “Frame Averaging for Invariant and Equivariant Network Design.”

**Limitations:**

The authors don't really state when their method is or is not preferred over conventional equivariant networks.

**Recommended Decision:**

3: Accept

**Relevance:**

3: Solid fit

**Strengths And Weaknesses:**

Strength:
- The proposed method is elegant.
- The paper is clearly written.

Weaknesses:
- I have concerns about the existence of an equivariant function $h:X \to G$. Such a function does not generally exist. For example, for any $x$ and any $g \in G$ such that $\rho(g)x=x$, $h(\rho(g)x)=h(x) \neq \rho(g)h(x)$. For linear representations, this is always the case for $x=0$. A more concrete example could be an image with some symmetry. Hence, it seems like the assumptions to theorem 1 are never met. This is not a big issue in practice, as typically only a measure 0 set has symmetries, and can be fixed by nuancing the claim in theorem 1, but this is important for the paper to be sound.
- The empirical evaluation on just rotated MNIST is very minimal.

**Submission Track:**

Proceedings Paper (9 Page)

---

### Official Review · Reviewer_ccWZ · 2022-10-14
**Promising idea, possibly high impact on real-world performance, some concerns on clarity and reproducibility**

**Confidence:** 3
**Soundness:** 3
**Presentation:** 2
**Contribution:** 3
**Overall Rating:** 6

**Summary:**

This submission proposes to split the construction of group-equivariant architectures into first a simple and not necessarily very expressive network equivariant by the group action by construction, together with a very expressive network without equivariance consideration. The method proposed consists in using the first small network as a cheap input-standardization procedure, thus granting the joint network equivariance, and experiments with rotated MNIST digits show that the total computational cost scales well with respect to the group size for comparable or improved performance.

**Questions:**

01. On page 13, why do you say that when $K$ is a normal subgroup of $G$, the group can be taken to be isomorphic to a semidirect product ? As far as I know, this is false. Is there an argument I am missing ? The standard counterexample is the group of Hamilton’s quaternions $\mathbb{H}_8 = \\{ \pm 1, \pm i, \pm j, \pm k\\}$ with usual quaternion multiplication, which is not a semi-direct product of subgroups.

02. Why does the conclusion list the optimization approach as future work ? It seems to me that the image algorithm of page 5, stacking feature maps and taking an argmax of activations, already corresponds to the optimization approach.

**Limitations:**

This work is considered preliminary, its limitations are not discussed by the authors, but the initial results are promising.

**Recommended Decision:**

3: Accept

**Relevance:**

4: Highly relevant

**Strengths And Weaknesses:**

The idea of normalizing inputs with a relatively simple procedure instead of focusing the work on the proof that certain classes of architectures are sufficiently expressive is very interesting, and the examples presented are promising. The main text is well written, and the connection to biological intuition is interesting, but the choice of vocabulary is often confusing, and the theoretical statements are not stated very clearly. On several occasions during this review, I found I needed to take a significant time to understand what was being stated by the authors, though the underlying ideas were relatively simple and not particularly surprising. There is also a mistake in the definition of the canonical mapping, whose severity in the applications I am unable to assess, for lack of implementation details on the choice of the $s$ map.

#### **On the use of the term “canonical”**

A major source of confusion during my reading has been the unorthodox use of the term “canonical” in this submission. This word has no clear definition in general and is used with different meanings in different fields, its usage is always associated with a lack of clarity to broad audiences, I don’t understand why it is given such a central place in this submission. As far as I know, the part everybody seems to agree on is that an object is canonical if it is “choice-free” essentially, that is to say, if two people given the same instructions would necessarily construct the same object.

For instance, for a finite-dimensional inner-product space $V$, a hermitian linear map $A : V \to V$ (i.e. satisfying $A^T=A$) induces a canonical ordered basis of eigenvectors (up to rescalings) if and only if all its eigenvalues have multiplicity one. If all eigenvalues are distinct indeed, one can perform the singular value decomposition of $A$, recover eigenvectors uniquely corresponding to each eigenvalue, which can be ordered by decreasing eigenvalue. There is no such way to identify a basis for the identity $I : V \to V$, and in general for any eigenspace of dimension superior to one, two people would in general choose different bases, since every rotation and rescaling is admissible, so there is no way to ensure a “unique” choice (although what counts as unique varies across fields).

In this sense, the “canonical” mappings presented here are not canonical, because they require a choice of representative which cannot be performed canonically (in the sense "without choosing arbitrarily"). It seems to me that “canonical” is used here to mean just a chosen unique representative per orbit, such that each orbit is mapped to its unique representative, without requiring the choice of representers to be unique. In this case, I don’t think the use of this term brings anything to the discussion, the procedure presented is just an input standardization, normalizing inputs to the orbit representative during preprocessing.

I think the use of the term “canonical” requires at least a short explanation in the introduction, a long footnote or a discussion in appendix, ideally a reference to an external source clarifying the meaning used here, just to make clear that the authors are not claiming that this choice of standardization is unique in any way, which would be a much stronger claim than what is presented here.


#### **Detailed comments (critical)**

01.  First and foremost, on the bottom of page 4, the claim “such a mapping is equivariant provided that $s : \mathcal{X} \to \mathbb{R}$
 has a unique minimum” is false. What is required, not only for equivariance but also for well-definition of $h$ above, is that $s$ admits a unique minimum on every orbit, which is significantly harder to enforce. The simplest counterexample that I can think of is $\mathcal{X} = \mathbb{C}$ and $s : x \mapsto \lvert x \rvert^2$, with the rotation group $G=S_1 \approx s^{-1}(1)$ acting by complex multiplication, $s$ has a unique minimum but is constant on orbits so $h$ remains undefined. It seems to me that for the applications, it is never checked that $h$ is well defined. This minimum not being unique, coupled with a machine-implemented argmin with an arbitrary choice, would mean that the resulting proposed network is not equivariant, which defeats the point of this work. It is crucial to enforce that the minimum of $s$ is unique on every orbit, and I could find no discussion of this in the rest of the paper.

02. On page 1, first paragraph, “the transformations with respect to which we require a model to be […] equivariant are known and provide a strong inductive bias” seems to be a contradiction. Either the equivariance is a requirement, and thus a constraint in the optimization, or it is not a constraint on the contrary and we are trying to “bias” towards more equivariant functions. It seems to me that the term “inductive bias” is misused here, either speak of restrictions / constraints or clarify what you mean by “inductive bias”, which quantity is being biased and with respect to what references.

03. The use of quantifiers in Theorem 1 is very unclear, leaving universal quantifiers at the end is acceptable if everything is interchangeable, but here Theorem 1 seems to read “$\exists k, \forall g, \forall x$" when it should be “$\forall g, \forall x, \exists k$". This problem keeps coming up throughout the paper, although it would be easily fixed by just defining variables in order. See for instance on Appendix A page 13, the proof of Theorem 1, with use of undefined variables $k′′$ and $k′′′$, although they could just be defined beforehand in closed form, which would make the proof much easier to read, and additionally make explicit their dependence on $x$.

04. On the top of page 4, there is a type problem in the sentence “the canonical mapping should output a coset in $G / K$”, the canonical mapping has values in $\operatorname{Im}(\rho)$ not in $G$. The whole sentence is not particularly clear, but the comment is interesting, and particularly important to clarify that there is an arbitrary choice to be made (cf. use of “canonical” above). The names “canonical mapping” and “canonical sample” are also somewhat confusing, in that despite what the names suggest, the canonical mapping h does not map samples to the canonical sample. Instead, said map taking samples to the canonical samples $(x \mapsto h(x)x)$ is not given an explicit name.

05. On page 5, equivariance with respect to $E_2$ is used without specifying the group action or the input space. Functions are invariant to a group action, not just a group, since a single group can admit several actions on the same space. Previously, it was more or less implicit that the action was induced by the general linear group through $\rho : G \to \operatorname{GL}(\mathcal{X})$, but in section 3.2 the setting has changed, the input space is now images, which are themselves viewed as functions, and the group acts by precomposition. This needs to be stated.

06. Related to the previous point, on the same paragraph of page 5, theorem 1 is used without checking its assumptions ($\rho$
is not specified, there is no check that the operator norm of $\rho$ is always bounded). A quick sentence to specify $\rho$ and the induced group action could mention that the boundedness is satisfied (no need for long proofs). Additionally, it is only at this point that it becomes really clear that the boundedness assumption is “$\forall \rho, \exists c$" rather than “$\exists c, \forall \rho$” as I had initially read, because the translation group satisfies exactly one of these conditions. This could have been clearer on page 3.

07. The proof of Theorem 2 in Appendix B is very loose on the use of quantifiers, and fails to mention that the constant $c$
can be chosen uniformly in $x$ (cf. previous point), for which the assumption that the input set is compact is critical, although it does not appear anywhere in the current proof.

08. On page 9, the use of the term “the correct canonical mapping” is incorrect, since there is no unique canonical mapping there can be no “correct” one.

09. There are various typos in the code: use of undefined ‘x’, argument ‘images’ unused, alternative use of ‘fibre_features’ or ‘fibres_activations’ (or 'fibre_activations', or ‘fibre_featuresr’ with an additional r). Adding a generation of a random image as example and running the code would have prevented these mistakes, and simplified understanding of it greatly.

10. There are no details on how the G-CNN are trained, which objective they minimize, or which frozen weights are used, which would make these results very hard to reproduce.


#### **Detailed comments (minor)**

11.  In section 2.3, just assume without loss of generality that $\mathcal{K}$ is closed under the group action (otherwise extend $\mathcal{K}$ to its closure for free). The current formulation makes it seem like this is a restrictive assumption although it is not.

12. On the top of page 5, “for practical application, we are typically interested in low-dimensional Lie groups” is not backed by references. At least name a few such applications if such references are not easy to list.

13. On page 6, section 3.2.2, “inductive bias” is used again, although it appears that what is meant is just “[This group captures the] prior knowledge”. It seems to me that there is no “bias” involved here.

14. On Table 1, the number of significant digits used in inconsistent and messes with the alignment.

15. On page 7, “a patience of 20 epochs” provides neither a reference nor an explanation (a short footnote would seem sufficient).

16. In figure 2, the radius of the disks is not described in the legend.

17. On page 9, there is an unclosed parenthesis in the first paragraph. The use of “they struggle” to refer to the networks is also a little odd.

18. On page 9, “learning the canonical network is a better approach” clearly overstates the results of the experiment. In particular, constructing by hand a unique representer for each orbit and implementing it explicitly should intuitively yield comparable results as learning the same function, but this is not present in experiments. The performance observed in the experiments is interesting, but the approach cannot be claimed better in general on the basis of this experiments alone.


**Submission Track:**

Proceedings Paper (9 Page)

---

### Official Review · Reviewer_dp4W · 2022-10-15
**Simple but effective idea to achieve equivariance, good results and theoretical underpinning**

**Confidence:** 4
**Soundness:** 4
**Presentation:** 4
**Contribution:** 4
**Overall Rating:** 9

**Summary:**

The paper describes an effective approach to achieve equivariance without working with specialed equivariant layers from begin to end. The idea is to predict a transformation (e.g. rotation) that can be used to align the input data to some canonical pose. After this alignment the remaining layers do not have to be equivariant anymore. This is interesting as in such an approach one can stick to light weight architecture whilst staying equivariant from begin to end, only the pose predictor (canonical map) has to be an equivariant architecture. The approach could possibly be used to scale up to larger symmetry groups for which equivariant architectures can be prohibitively expensive. The method is theoretically well underpinned and the paper contains a universal approximator result for the proposed architecture. The experimental give a good impression of the benefits of the method, which is efficiency + equivariance.

**Questions:**

It was to me unclear what the experiment with "frozen weights" encompasses. Does this mean random initialization of the canonical map network and then don't train it?

**Limitations:**

I think the paper is clear, though perhaps the practical issue of implementing the paper for arbitrary transformation groups can be addressed.

**Recommended Decision:**

3: Accept

**Relevance:**

4: Highly relevant

**Strengths And Weaknesses:**

**Strenghts**

* The paper presents an intuitive idea that is very well executed.
* The paper is also very well written and nicely motivated from the neuroscience perspective.
* The paper is precise, although some parts were more technical than others, all details were there which allowed me to confirm the soundness.
* Although only applied on MNIST, the experiments provide valuable insights on performance (both in accuracy and speed) via well designed ablations.

**Weaknesses**

* It would have been nice to see the method applied to a more challenging dataset then MNIST.
* The paper claims that the method allows for implementation for arbitrary transformation groups. This is theoretically correct I suppose, but practically a major challenge to this beyond translations, rotations and scale. Though there are methods out there for implementations for arbitrary Lie groups [1,2] and recently also for translation+rotation+scale [3].
* The method is based on the straight-through trick to make the argmax differentiable. Perhaps a more stable approach would be to convert the orientation fiber to a one that transforms via irreps of frequency one via a Fourier tranform. This would effectively give a direction vector which can be turned into a rotation matrix as explained in the paper already. The idea would then be to work with *steerable* group convolutional neural networks as to predict vectors [4].

[1] Bekkers, E. J. (2019, September). B-Spline CNNs on Lie groups. In International Conference on Learning Representations.

[2] Finzi, M., Stanton, S., Izmailov, P., & Wilson, A. G. (2020, November). Generalizing convolutional neural networks for equivariance to lie groups on arbitrary continuous data. In International Conference on Machine Learning (pp. 3165-3176). PMLR.

[3] Knigge, D. M., Romero, D. W., & Bekkers, E. J. (2022, June). Exploiting redundancy: Separable group convolutional networks on lie groups. In International Conference on Machine Learning (pp. 11359-11386). PMLR.

[4] Cesa, G., Lang, L., & Weiler, M. (2021, September). A Program to Build E (N)-Equivariant Steerable CNNs. In International Conference on Learning Representations.

**Submission Track:**

Proceedings Paper (9 Page)

---

### Decision · Program_Chairs · 2022-10-21

Accept (Oral)